# Challenges and Therapeutic Opportunities in the dMMR/MSI-H Colorectal Cancer Landscape

**DOI:** 10.3390/cancers15041022

**Published:** 2023-02-06

**Authors:** Núria Mulet-Margalef, Jenniffer Linares, Jordi Badia-Ramentol, Mireya Jimeno, Carolina Sanz Monte, José Luis Manzano Mozo, Alexandre Calon

**Affiliations:** 1Institut Català d’Oncologia, 08916 Badalona, Spain; 2Hospital del Mar Medical Research Institute (IMIM), 08003 Barcelona, Spain; 3Hospital Germans Trias i Pujol, 08916 Badalona, Spain

**Keywords:** colorectal cancer, mismatch repair, microsatellite instability, immunotherapy

## Abstract

**Simple Summary:**

Between 5% and 15% of colorectal cancers (CRC) show deficiencies in the mismatch repair machinery and high microsatellite instability (dMMR/MSI-H). dMMR/MSI-H CRC is characterized by a dysfunctional DNA repair system, which renders the tumor immune microenvironment more susceptible to immunotherapy. Currently, immunomodulating drugs are included in the therapeutic arsenal of dMMR/MSI-H CRC and have substantially improved cancer treatment. However, an important proportion of patients with dMMR/MSI-H CRC show primary or acquired resistance to immunotherapy due to molecular traits that are yet to be fully elucidated. Here, we review the current understanding of dMMR/MSI-H CRC molecular and clinical features and discuss their therapeutic implications for CRC patients.

**Abstract:**

About 5 to 15% of all colorectal cancers harbor mismatch repair deficient/microsatellite instability–high status (dMMR/MSI-H) that associates with high tumor mutation burden and increased immunogenicity. As a result, and in contrast to other colorectal cancer phenotypes, a significant subset of dMMR/MSI-H cancer patients strongly benefit from immunotherapy. Yet, a large proportion of these tumors remain unresponsive to any immuno-modulating treatment. For this reason, current efforts are focused on the characterization of resistance mechanisms and the identification of predictive biomarkers to guide therapeutic decision-making. Here, we provide an overview on the new advances related to the diagnosis and definition of dMMR/MSI-H status and focus on the distinct clinical, functional, and molecular cues that associate with dMMR/MSI-H colorectal cancer. We review the development of novel predictive factors of response or resistance to immunotherapy and their potential application in the clinical setting. Finally, we discuss current and emerging strategies applied to the treatment of localized and metastatic dMMR/MSI-H colorectal tumors in the neoadjuvant and adjuvant setting.

## 1. Introduction

Colorectal cancer (CRC) is one of the leading forms of malignant tumors worldwide with estimated rising incidence in the following decade [1]. Comprehensive analysis of CRC molecular biology has defined common features of patients with CRC that are used for clinical decision-making, such as *RAS* mutational status, BRAF V600E mutations, and hypermutant phenotypes, among others [2]. Of note, the hypermutant phenotype referred to as microsatellite instability high (MSI-H) status is associated with the inactivation of mismatch repair (MMR) genes resulting in MMR deficiency (dMMR). MMR genes *MLH1*, *MSH2*, *MSH6,* and *PMS2* effectively control and correct errors introduced in short repeated sequences of 1–6 nucleotides and are widely distributed in the DNA, mostly near coding regions. These sequence motifs, also called microsatellites, are prone to polymerase slippage during replication, resulting in a higher rate of sequence alterations (i.e., loss or insertion of repeat units) [3]. Consequently, an impaired MMR machinery in dMMR/MSI-H tumors results in an increased mutational rate compared to MMR proficient/microsatellite stable (pMMR/MSS) CRC. Indeed, dMMR/MSI-H CRC commonly display 10 to 100-fold greater number of somatic mutations compared to pMMR/MSS tumors [4].

dMMR/MSI-H is a pan-tumor phenotype representing nearly 15% of all CRCs. It was firstly reported in inherited cancers associated with Lynch syndrome and was later described in sporadic CRC [5,6,7]. Yet, the dMMR/MSI-H phenotype occurs mostly in sporadic CRC through *MLH1* promoter methylation [5,7,8]. dMMR/MSI-H CRCs have a distinct pathological profile including right-sided primary, mucinous, and poorly differentiated tumors as well as increased occurrence of *BRAF* mutations [9]. Approximately 20% of stage II, 12% of stage III, and 4% of stage IV CRC tumors are diagnosed as dMMR/MSI-H, thus suggesting an association of dMMR/MSI-H with earlier stages of CRC and with better prognosis [10,11]. This favorable outcome is likely related to the generation of large amounts of neoantigens associated with the mutational rate of dMMR/MSI-H tumors, which results into an immunogenic tumor microenvironment (TME) [12,13]. In contrast, retrospective series and pooled analyses pointed out the dMMR/MSI-H phenotype as a negative prognostic factor in the metastatic setting, potentially due to the coexistence of BRAF V600E mutation in around one third of cases [11,14]. However, these data were analyzed before the irruption of immunotherapy, which entailed profound changes in the natural evolution of advanced dMMR/MSI-H CRC and has thus become a new standard in this niche [15]. Even so, further research is needed to define the patterns of benefit or resistance to immunotherapy in dMMR/MSI-H CRC.

## 2. The Clinical Assessment of dMMR/MSI-H Phenotype

### 2.1. Microsatellite Instability Detection by Fluorescent Multiplex PCR

MSI testing is commonly performed by multiplex PCR using fluorescent primers and followed by capillary electrophoresis fragment analysis. The Bethesda revised microsatellites markers (commonly mononucleotide *BAT-25*, *BAT-26*, *NR-21*, *NR-4* and *NR-27*/*MONO-27*) are the recommended PCR targets for the characterization of MSI/MSS status due to their stability between individuals [16,17]. Accordingly, a tumor is considered MSI-H when the size of two or more of these markers is altered, whereas only one or no alterations at all are characteristic for MSS status. Of note, new targets are continuously being evaluated. Among them, *ACVR2A*, *BTBD7*, *DIDO1*, *MRE11*, *RYR3*, *SEC31A* and *SULF2* have been identified through genome sequencing of different MSI-H tumors [18]. These markers, selected for their applicability in different ethnicities and over cancers of distinct origins, are currently available for clinical testing [19,20].

### 2.2. Immunodetection of MMR Proteins

Immunohistochemical (IHC) detection of MMR proteins MLH1, PMS2, MSH2, and MSH6 is widely used in the clinical setting to assess the microsatellite status, mainly because of its robustness and the straightforward interpretation of results by well-trained pathologists [21]. Similar to MSH2 and MSH6, MLH1 and PMS2 form a heterodimer that recognizes and binds to mismatch base pairs. Therefore, the concurrent loss of both proteins is frequently detected in dMMR [4]. Furthermore, the detection of this panel of MMR-related biomarkers provides mechanistic insights on the dMMR/MSI-H phenotype [22]. The detection of all four MMR proteins expression defines proficient mismatch repair system (pMMR). In contrast, dMMR is characterized by the lack of expression of at least one of these markers [23].

### 2.3. Limitations of the Clinical Assessment of dMMR/MSI-H Status

Although dMMR and MSI-H status are mostly equivalent, some exceptions have been reported. For instance, dMMR caused by *MSH6* germline mutation does not meet the criteria of MSI-H diagnosis [24]. Conversely, IHC does not segregate functional and nonfunctional proteins, thus limiting the diagnostic power of this approach. For instance, approximately 5% of the alterations observed in MMR genes are due to missense mutations (especially in *MLH1*) that lead to the expression of functionally impaired proteins. Therefore, the detection of MLH1, PMS2, MSH2, and MSH6 expression does not entirely exclude microsatellite instability. In this regard, it is worth noting that positive IHC staining, while being either homogeneous or heterogeneous, will be commonly considered as “intact” expression by clinical pathologists [25]. Staining positivity is usually contrasted with positive reaction in internal control cells (nuclei of stromal, inflammatory, or non-neoplastic epithelial cells). However, heterogeneous staining may challenge the clinical evaluation of MMR status since no clear guidelines are available to date [25,26,27]. In this line, misdiagnosis/failure to identify either MSI or dMMR is not rare. For instance, the assessment of MSI or dMMR status in patients with metastatic CRC associates with 5–10% false-positive cases [28]. Therefore, the sequential application of IHC followed by PCR is recommended to reduce incompatibility and minimize false negative results [20].

### 2.4. New Strategies for the Evaluation of dMMR/MSI-H Status

With the advent of next generation sequencing (NGS), new options are being developed to evaluate more accurately MSI status [18,29,30]. In this line, methods such as whole genome/exome sequencing or targeted gene sequencing allow the detection of alterations in thousands of microsatellite loci. MSI status assessment by NGS relies on computational algorithms able to eliminate the bias introduced by polymerase slippage [31]. In this line, MSIsensor was one of the first algorithms developed in paired tumor and matched normal samples [32]. In contrast, mSINGS is based on the comparison of tumor samples with normal population controls as baseline [30]. More recent and accurate algorithms include MOSAIC [29], MANTIS [33], the Cortes-Ciriano method [34], mSILICO [35], and MSINGB [36]. Alternatively, radiomics is emerging as a promising tool for the MSI diagnosis. Ensemble models based on PET/CT images have been constructed based on this approach, combining clinical risk factors and machine learning-based analysis models. Although still under development, this technique may provide a quantitative, efficient, and non-invasive method for the assessment of MSI status in the near future [37].

## 3. Molecular Features of dMMR/MSI-H CRC

### 3.1. Intrinsic Characteristics of dMMR/MSI-H Cancer Cells

Seminal studies in CRC have pointed out the molecular idiosyncrasy of dMMR/MSI-H CRC (Figure 1). Exome sequencing analyses from The Cancer Genome Atlas (TCGA) project concluded that dMMR/MSI-H CRC, unlike pMMR/MSS tumors, is hypermutated, with a tumor mutational burden (TMB) above 12 mutations/Mb [2]. The TMB threshold was defined by the TCGA project itself, since there is no exact value that defines hypermutated tumors per se. In addition to the dMMR/MSI-H phenotype, mutations in *POLE* and *POLD1* genes can also promote a hypermutated phenotype [2].

Frameshift mutations are one of the hallmarks of dMMR/MSI-H CRC. Of note, genes that include microsatellites in their coding areas are particularly vulnerable to frameshift mutations, which may appear at very early stages of tumor development in dMMR/MSI-H CRC. As a result, insertions or deletions of dinucleotides or trinucleotides in microsatellites are not repaired, which leads to the generation of highly immunogenic neoantigens [31]. Indeed, an in silico pan-tumor study using exome sequencing data from the TCGA showed that frameshift mutations generated almost three times more neoantigens that are recognized by T cells compared to any other non-synonymous mutations [38].

The study of the mutational profile in dMMR/MSI-H CRC is largely intricate due to the complexity of its hypermutability status [39]. Yet, massive genomic sequencing analyses have elucidated the most frequent molecular aberrations affecting the dMMR/MSI-H phenotype [39]. In particular, mutations in *TP53* appear in 20% of dMMR/MSI-H CRC compared to a 60% of pMMR/MSS CRC, and *APC* in 51% compared to 81%, respectively [2]. Mutations in KRAS and BRAF V600E are among the most frequent mutations in advanced stages of dMMR/MSI-H CRC, and they are each found in approximately 30% of the cases compared to frequencies of around 40% and 8%, respectively, of pMMR/MSS CRC cases [2,40]. Importantly, BRAF V600E mutation is unique to sporadic dMMR/MSI-H CRC, and exclusive to the presence of Lynch Syndrome. Indeed, pre-clinical data suggest that BRAF V600E mutations promote dMMR/MSI-H phenotype through the activation of the MAPK pathway. In turn, MAPK signaling induces the expression of MAFG, a transcriptional repressor that binds to *MLH1* promoter and recruits methyltransferases, resulting in promoter hypermethylation and inactivation [41]. *ZBTB2* is another gene mutated in dMMR/MSI-H CRC. Mutations in *ZBTB2* enhance the expression of MDM2, which regulates p53 degradation thus blocking its DNA repair effect [42,43]. Additional mutations are reported in the tumor suppressor *RANBP2*; in the mediator of p53 inhibition *PSRC1*; in *MECOM*, which encodes a transcriptional regulator of hematopoiesis, apoptosis, and proliferation; and in WNT pathway-related *RNF43* [42,43]. Genes that associate with antigen presentation can also be mutated in dMMR/MSI-H CRC, although with variable frequency. These genes include the transcriptional regulators of MHC type I *NLRC5* and *RFX5*; *TAP1* and *TAP2* involved in antigenic processing; as well as *HLA-A*, *HLA-B*, *HLA-C*, and *B2M* involved in antigen presentation [44]. In addition, genes encoding for subunits of epigenetic regulatory complexes, such as the SWI/SNF complex, also show a higher mutational incidence in dMMR/MSI-H tumors compared to pMMR/MSS [45].

### 3.2. Extrinsic Features of dMMR/MSI-H Tumors

In an effort to functionally stratify CRC tumors, Guinney and colleagues reconciled previous molecular classifications of CRC based on tumor transcriptomic profiles into four consensus molecular subtypes (CMS1 to 4). Remarkably, around 70% of dMMR/MSI-H CRCs are clustered in CMS1 [46], which is characterized by a strong influence of the TME. Consequently, this association indicates that dMMR/MSI-H CRC may be determined through their specific interactions with the TME in addition to the cancer cells intrinsic properties. Further transcriptomic analysis of the immune microenvironment revealed that CMS1 is also defined by genes associated with antigen presentation, interferon γ (IFNg) signaling, the T helper (Th) 1 phenotype, and chemoattractant cytokines such as CXCL9, CXCL10, and CXCL16 [47,48,49]. Accordingly, CMS1-dMMR/MSI-H CRCs exhibit higher abundance of immune cells in the tumor center and at the invasive margin, mainly consisting of cytotoxic T lymphocytes, natural killer (NK) cells, Th-1 and Th-2 lymphocytes, and M1-like macrophages [47,49,50]. Of note, the density of cytotoxic T-lymphocyte infiltration correlates with the number of frameshift mutations, thus indicating that lymphocytes may preferentially react to tumors with a higher frequency of neoantigen generation [51]. As a result, immunotherapies, particularly immune checkpoint inhibitors (ICI), have shown striking positive effects in dMMR/MSI-H mCRC [52,53].

Notwithstanding their favorable immune landscape, dMMR/MSI-H CRCs are also particularly prone to raise adaptive resistance to anti-cancer immunity (Figure 1). For instance, a dMMR/MSI-H status is also associated with increased expression of suppressive immune checkpoints such as indolamine 2,3-dioxygenae (IDO-1), cytotoxic T-lymphocyte antigen 4 (CTLA-4), programmed death receptor (PD-1) and its ligand (PD-L1), and lymphocyte activation gene 3 (LAG-3), among others [13,49,54]. In addition, recent studies have shown that a subset of dMMR/MSI-H is intrinsically characterized by low cytotoxic lymphocyte infiltration [55] or by increased abundance of immunosuppressive myeloid cells and fibroblasts [56] (Figure 1). These scenarios could be the result of similar activation of WNT or MAPK pathways to pMMR/MSS CRC, which lead to immune desertion [57].

### 3.3. Confounding Subsets of dMMR/MSI-H and pMMR/MSS CRC

In contrast to dMMR/MSI-H, pMMR/MSS CRCs are considered as immune excluded tumors. However, around 10% of pMMR/MSS tumors display an immunogenic profile similar to dMMR/MSI-H CRCs [58]. This was observed in the CCTG CO.26 trial, in which patients with refractory pMMR/MSS CRC were randomized to receive anti-PD-L1 (Durvalumab) and anti-CTLA-4 (Tremelimumab) monoclonal antibodies (mAbs) versus best supportive care. In this setting, patients with TMB above 28 mutations/Mb (21% of patients) experienced greater benefit from immunotherapy [59]. Remarkably, these high TMB levels were similar to the ones observed in dMMR/MSI-H tumors, thus suggesting that a subset of pMMR/MSS CRCs harbor hypermutant phenotype either per se or as a result of mutational footprints induced by chemotherapy [60]. Indeed, transcriptional analysis indicated that pMMR/MSS CRC responding to immunotherapy displayed a relative abundance of cytotoxic lymphocytes similar to dMMR/MSI-H CRC [61].

Conversely, a shift from pMMR/MSS to dMMR/MSI-H phenotype has been observed upon treatment with anti-EGFR mAbs and BRAF inhibitors in preclinical models of CRC [62]. This suggests that dMMR/MSI-H status is potentially acquired upon therapy as an adaptive mechanism of drug resistance. Along this line, alquilating agents such as Temozolamide are also able to induce hypermutational phenotype in pMMR/MSS CRC. Their combination with immune checkpoint inhibitors is currently evaluated in clinical trials as a promising strategy to sensitize unresponsive tumors to immunotherapy [63,64].

An increasing body of literature has reported a heterogeneous expression of MMR proteins within the same tumor, thus suggesting the existence of a MSS/MSI heterogeneous (MSS/MSI-het) subset of CRC [25,26,27]. These peculiar patterns are not artefactual since focal or zonal losses of MMR protein expression consistently correlate with MSI-H-associated molecular traits in the corresponding areas [26,27]. Even though this mixed MSS/MSI-het phenotype may be limited to a subset of CRCs, the coexistence of pMMR/MSS and dMMR/MSI-H malignant cells raises important questions that are not addressed by the current guidelines. For instance, the relative abundance of dMMR/MSI-H and pMMR/MSS cancer cells in a given tumor may both have an impact on diagnosis, treatment decision-making, and ultimately patient benefit from therapy. Yet, the penetrance of this MSS/MSI-het phenotype remains largely unknown and additional studies are still needed to elucidate its implication with CRC outcome.

## 4. Therapeutic Implication of dMMR/MSI-H Status

### 4.1. Initial Studies and Treatment in the Metastatic Setting

Early studies on the overall population of mCRC patients showed a very limited clinical benefit from ICIs. In a phase I trial that included 39 refractory solid tumors, only 1 patient presented a complete pathological response that lasted for more than 3 years. Interestingly, this mCRC patient displayed dMMR/MSI-H status [65]. In KEYNOTE-028 phase I trial, treatment with anti-PD-1 mAbs (Pembrolizumab) was evaluated in 23 solid tumors. One partial but long-lasting response was recorded. Here again, dMMR/MSI-H status was associated to this mCRC patient [66]. In line with these observations, the phase II KEYNOTE-016 trial evaluated the clinical efficacy of Pembrolizumab in patients with pMMR/MSS mCRC, dMMR/MSI-H mCRC, and dMMR/MSI-H non-colorectal cancer. A clear lack of response to therapy was observed in the cohort of 18 patients (0% ORR) displaying pMMR/MSS status. In contrast, the overall response rate (ORR) reached 40% in the 10 patients harboring dMMR/MSI-H mCRC. The updated results have shown a similar evolution, in which the disease control rate (DCR) and ORR were 89% and 50%, respectively, for dMMR/MSI-H mCRC (n = 28) versus a 16% and 0% for pMMR/MSS mCRC (n = 25). PFS and OS after 9 months median follow-up was not reached for dMMR/MSI-H mCRC compared to 2.4 and 6 months, respectively, for pMMR/MSS mCRC [67]. Pembrolizumab’s robust anti-tumor activity in mCRC was confirmed in the phase II KEYNOTE-164 in second or subsequent lines of treatment [68]. A similar positive effect was observed in the phase II Checkmate 142 trial, which evaluated the efficacy of another anti-PD-1 mAb (Nivolumab) in 74 chemorefractory patients with dMMR/MSI-H mCRC. Strikingly, PFS and OS at 12 months reached 50% and 73%, respectively [53]. Based on these promising results, US FDA approved Pembrolizumab and Nivolumab as the second-line treatment for dMMR/MSI-H mCRC in 2017. Since then, Pembrolizumab was evaluated in a phase III trial in dMMR/MSI-H mCRC as first line of treatment (KEYNOTE-177). Again, Pembrolizumab significantly improved both PFS and OS when compared to standard chemotherapy. In addition, the rates of Grade 3–5 treatment-related adverse events were 22% for Pembrolizumab (colitis, hepatitis) and 66% for chemotherapy (neutropenia, diarrhea, fatigue), thus indicating a reduced toxicity of the treatment [69].

It is worth mentioning that dual ICI regimen demonstrated even more benefit for dMMR/MSI-H CRC patients. For example, the phase II Checkmate 142 trial that included a cohort of 119 pretreated patients with dMMR/MSI-H mCRC showed an ORR of 55% and a 12-months OS of 83% upon anti-CTLA-4 (Ipilimumab) and Nivolumab dual therapy [70]. The same combination was administered as first line treatment to 45 patients with dMMR/MSI-H mCRC. In this setting, the ORR and OS at 12 months reached 60% and 83% respectively [71], thus suggesting that the dual inhibition of CTLA-4 and PD-1 synergistically promotes anti-tumor responses.

Currently, a great body of clinical research is exploring the potential benefit of combining ICIs with systemic and/or targeted therapies. In this line, the Checkmate 8HW (NCT04008030) phase III trial is currently evaluating the efficacy of Nivolumab monotherapy, Nivolumab plus Ipilimumab, or chemotherapy in patients with dMMR/MSI-H mCRC. On the other hand, preclinical data suggested that VEGF blockade induces an immune permissive TME improving the benefit from ICI [72,73,74]. On the basis of these results, the combination of ICI-anti-VEGF/VEGFR mAbs plus chemotherapy was evaluated in early clinical trial in dMMR/MSI-H mCRC. Treatment was well tolerated and associated with positive clinical activity [75]. Following on this, the COMMIT phase III trial (NCT02997228) is now evaluating the combination of anti-PD-L1 mAbs (Atezolizumab) with chemotherapy (FOLFOX) and anti-VEGF mAbs (Bevacizumab) in the first line treatment of dMMR/MSI-H mCRC [76]. Additionally, a single center phase II trial conducted in Chinese population is assessing chemotherapy (FOLFIRI)-Bevacizumab combination plus anti-PD-1 mAbs (Nivolumab or Pembrolizumab) as second-line therapy in dMMR/MSI-H mCRC (NCT05035381) [77]. Results from pivotal trials in patients with dMMR/MSI-H are summarized in Table 1.

### 4.2. Adjuvant and Neoadjuvant Therapy in Localized CRC

While only ~5% of mCRC are dMMR/MSI-H, this subset represents about 10–12% of stage III CRC [78]. Consequently, the good results obtained from ICI-based therapies in dMMR/MSI-H mCRC have led to their evaluation in patients with non-metastatic disease. Two trials intended to test the potential efficacy of ICI as adjuvant treatment have been designed so far. On the one hand, ATOMIC phase III trial is evaluating the combination of mFOLFOX6 plus Atezolizumab compared to mFOLFOX alone (NCT02912559) [79]. On the other hand, POLEM trial aims to assess Avelumab (anti-PD-L1 mAb) treatment following adjuvant 5-FU based chemotherapy in dMMR/MSI-H and/or POLE exonuclease domain mutant CRC tumors [80]. However, while ATOMIC is currently recruiting patients, POLEM trial encountered significant technical challenges and was recently discontinued. 

Additionally, the potential of ICI-based regimen for dMMR/MSI-H CRC is also being explored in the neoadjuvant setting. In this regard, the NICHE phase II trial assessed the efficacy of short course neoadjuvant treatment with Nivolumab and Ipilimumab. All 21 treated CRCs showed either major or complete pathological response. In contrast, only three major pathological responses were observed in the 15 pMMR/MSS CRCs [81]. In addition, a recent prospective phase II evaluated anti-PD-1 mAbs (Dostarlimab) as neoadjuvant treatment for dMMR/MSI-H stage II-III rectal adenocarcinoma. Strikingly, all 12 patients maintained clinical and radiological complete response during the 6-month follow-up [82]. Recently, the results of the NICHE-2 study were recently presented at the ESMO 2022 edition in Paris. In this study, neoadjuvant treatment with a cycle of Nivolumab plus Ipilimumab followed by another cycle with Nivolumab prior to surgery was assessed in 107 patients with locally advanced colon tumors (Stage II-III), with the objective of studying safety and response rate. With 13 months of median follow-up, a major pathological response rate (less than 10% residual viable tumor) of 95% was achieved, with a 67% of complete responses (0% residual viable tumor).

The use of immunotherapy as neoadjuvant treatment in CRC shows a great potential as a therapeutic strategy. However, there is still a need for more clinical studies to evaluate their benefits over current adjuvant strategies. Firstly, few clinical trials are evaluating the safety of immunotherapy and the possible surgical complications related to neoadjuvant treatment. Secondly, more efforts are needed to assess the effect of immunotherapy in the neoadjuvant setting. In this context, it remains unclear whether selected patients should follow an organ-preserving approach or a “watch and wait” strategy. Finally, the combinations of neoadjuvant immunotherapy with radiotherapy and/or chemotherapy, as well as the proper timing of incorporation (concomitant or sequential), should be investigated and optimized.

### 4.3. New Directions

Even though ICIs have demonstrated encouraging results, a substantial proportion of dMMR/MSI-H CRC patients still do not benefit from current treatment options. This calls for a more personalized therapy based on additional/original molecular markers. For example, while Pembrolizumab is indicated as first line therapy in unresectable dMMR/MSI-H mCRC, Encorafenib, a BRAF inhibitor used in combination with Cetuximab (anti-EGFR mAb) is only indicated in second line therapy for BRAF V600E-mutated mCRC patients. Therefore, and in the absence of established first line therapy, SEAMARK (NCT05217446) phase II trial was initiated in 2022 to evaluate the combination of Encorafenib, Cetuximab, and Pembrolizumab as first line treatment in in dMMR/MSI-H and BRAF V600E-mutated mCRC patients [83].

Alternatively, evidence indicates that dMMR/MSI-H tumors are enriched in NTRK fusions [84]. Indeed, Coco and colleagues identified up to 15% of dMMR/MSI-H mCRC harboring kinase fusions [85]. Remarkably, a recent study of dMMR/MSI-H mCRC patient with NTRK fusion treated with Larotrectinib, an inhibitor of tropomyosin kinase receptors TrkA, TrkB, and TrkC [86,87,88], reported a striking therapeutic response after ICI failure [89]. This suggests Larotrectinib as a potential therapeutic strategy in ICI-resistant dMMR/MSI-H tumors. Additional molecules inducing T cell activation or blocking T cell checkpoint inhibitors are currently evaluated in early clinical trials involving dMMR/MSI-H CRC patients [90]. In this line, studies have highlighted the expression of additional immune checkpoints markers such as LAG-3, T cell immunoglobulin mucin domain 3 (TIM-3), and IDO-1 in the MSI population, thus making it a rationale for the use of different ICIs [91]. Indeed, anti-TIM3 in combination with anti-PD-L1 mAbs showed promising clinical activity in dMMR/MSI-H tumors of distinct origins [92]. Therefore, the identification of new actionable targets may provide original treatment opportunities for dMMR/MSI-H CRC patients unresponsive to current ICI.

## 5. Predictive Factors of Therapeutic Outcome

### 5.1. Clinical Factors

As mentioned above, the important effect of ICIs in the clinical setting led to their approval as second-line treatment for dMMR/MSI-H mCRC. However, about 30–50% of dMMR/MSI-H mCRC patients still display intrinsic resistance to immunotherapy [53,69,93]. Hence, the identification of predictive biomarkers of response remains crucial to guide therapeutic decision-making [94]. In this sense, several studies have identified potential clinical predictors of response to ICIs. Among them, Pietrantonio and colleagues developed a nomogram that integrates five clinical variables to estimate the outcome of dMMR/MSI-H mCRC patients receiving ICIs [95]. The use of a multivariable model provides a dual scoring system for 12-month PFS and for time-independent event-free probability (EFP) [95]. However, a prospective validation of these methods is still required to establish their relevance in the clinical setting.

Clinical presentation has also been associated with ICIs treatment outcome. In this context, Fuca et al. retrospectively evaluated a cohort of patients with dMMR/MSI-H gastrointestinal cancers treated with anti-PD-1 ± anti-CTLA-4 agents to determine whether malignant ascites impacted treatment outcome [96]. Authors showed that peritoneal metastases manifested with ascites associate with unfavorable outcome upon anti-PD-1 mAbs treatment, possibly due to a link between malignant ascites with an immunosuppressive microenvironment [97].

### 5.2. Microbiome

Different studies have suggested that the administration of broad-spectrum antibiotics (ATBs) have a negative impact on ICI treatment outcome [98]. ATBs may induce intestinal dysbiosis, altering the normal gut microbiota with adverse consequences on the immune system [99], and it may take up to 3 months for the gut microbiota to recover [100]. In this context, the microbiome has emerged as a potential biomarker of immunotherapy effectiveness. For instance, increased abundance of *Akkermansia muciniphila*, *Ruminoccoccus* sp. and *Faecalibacterium* sp. have been observed in fecal samples from melanoma patients responding to anti-PD-1/PD-L1 or anti-CTLA-4 mAbs [101,102]. *Fusobacterium nucleatum* are other bacteria with potential immunomodulating properties (Figure 1). According to preclinical data in CRC cell lines, *Fusobacterium nucleatum* enhance oncogenic events such as WNT or MAPK pathway activation precluding immunogenic activation [103,104]. Alternatively, *Fusobacterium nucleatum* are also able to bind to negative regulatory immune checkpoints like TIGIT and CEACAM1 [105,106], thus leading to reduced CD4^+^/CD8^+^ T cells infiltration [107]. It is worth mentioning that a tumor-associated fecal bacterial profile has not been described for dMMR/MSI-H CRC in light of the benefit from immunotherapy. Yet, reports indicate that *Fusobacterium nucleatum* are particularly enriched in these tumors [108,109] and preliminary results point to improved response to immunotherapy in the presence of *Fusobacterium nucleatum* [110].

### 5.3. Molecular Factors

The overall improved responses of dMMR/MSI-H colorectal tumors to immunotherapy results largely from an increased immune infiltration compared to pMMR/MSS tumors [13,91]. As mentioned above, acquired mutations in the MMR machinery leads to increased TMB, resulting into a vigorous tumor immune microenvironment. Consequently, TMB is often used as a clinical biomarker of response to immune checkpoint inhibitors [111]. However, since a proportion of patients with a high mutational density are refractory to current immunotherapeutic strategies, TMB alone might not an accurate predictor of immunologic responses. Therefore, it is worth understanding additional molecular and cellular escape mechanisms to refine better biomarkers for clinical decision-making. 

Mutations of key genes involved in immunological processes can be used as actionable biomarkers of therapeutic response in immuno-oncology, which could be relevant for dMMR/MSI-H tumors given their increased mutational rate. Of note, the elevated selective pressure posed by the immune surveillance often leads to damages in the MHC class I antigen presentation machinery [112,113,114]. Mutations of *B2M*, followed by the *HLA-B* heavy chain and *HLA-C*, are the most common alterations acquired by dMMR/MSI-H tumors preventing antigen presentation, and have been associated with negative responses to immunotherapy [112,114,115]. However, additional clinical studies have shown that mutations of *B2M* do not associate with resistance to immune checkpoint inhibition, as dMMR/MSI-H tumors with *B2M* alterations can respond to immunotherapy [116,117]. Therefore, the use of the MHC I antigen presentation machinery as a clinical biomarker remains controversial. Alternatively, resistant dMMR/MSI-H tumors may acquire additional alterations that ultimately govern immune cells infiltration and activity. For instance, mutations in JAK1, a kinase downstream the IFNg pathway, leads to a decreased anti-tumor immune activity. Consequently, these tumors could diminish the overall expression of PD-L1, which in turn renders them more resistant to anti-PD-L1 targeted therapies [118,119,120,121]. Conversely, BRAF mutated dMMR/MSI-H tumors seem to be associated with increased infiltration CD8 cytotoxic T cells infiltration and antigen-presenting cells such as dendritic cells and M1-like macrophages [122,123,124]. Nevertheless, neither BRAF V600E nor RAS mutations have been meaningfully associated with benefit to immunotherapy in the aforementioned clinical trials. [53]

Overall, tumors with a low presence of an immune tumor microenvironment, also named “cold tumors”, are correlated with poor responses to immunotherapy. In this regard, it is currently accepted that high levels of TGF-beta and stroma impair cytotoxic immunological responses in pMMR/MSS tumors [125,126,127,128]. However, a proportion of dMMR/MSI-H tumors may also contain a TGF-beta enriched tumor microenvironment, which could result into similar mechanisms of immune evasion [46,129] (Figure 1). On the other hand, recent advances in RNA sequencing techniques, particularly at the single cell level, and multiplex fluorescent labelling highlight the quality, rather than quantity, of immune infiltrates as a predictive biomarker for the clinical practice. 

On the transcriptional level, Lal and colleagues defined in 2015 four categories of coordinate immune response clusters. In particular, cluster A, which is correlated with better responses to immunotherapy, is enriched in dMMR/MSI-H tumors [130]. In 2016, Becht and colleagues published the Microenvironment Cell Populations-counter (MCP-counter) method, which allows the quantification of stromal and immune cell populations using transcriptomic signatures [131]. A higher abundance of cytotoxic T cells as predicted by the MCP-counter is associated with improved prognosis in MSI CRC, and could be used to predict responses to ICI [132]. Recent research has pointed out several gene expression signatures associated with responses to ICIs, including CMS transcriptomes, IFNg signaling, and genes belonging to the cancer-immunity-cycle [57,133,134]. Alternatively, there is a yet nascent interest in the metabolomic biomarkers for CRC. Indeed, dysregulated metabolite markers including fatty acids, amino acids, and lysophosphatidylcholines have been associated with clinicopathological features of CRC, including therapeutic outcome [135,136,137]. Of note, and as recently discussed by Holbert and colleagues, the polyamine metabolism could potentially predict responses to ICIs [138].

On the histological level, seminal studies led by Jerome Galon have defined that the Immunoscore, a scoring system of immune cells with anti-tumor activity, robustly correlates with improved prognosis [139,140]. Interestingly, a study led by Mlecnik and colleagues shows that the Immunoscore outperforms the MSS/MSI status in predicting CRC outcome, and that a proportion of dMMR/MSI-H tumors show low Immunoscore values [47]. In this regard, a recent study by Pelka and colleagues suggested the existence of immunologic hubs that are shared between pMMR/MSS and dMMR/MSI-H colorectal tumors [56]. Supporting the evaluation of immune infiltrates and activity as prognostic predictors, Corti and colleagues proposed a Pan-Immune-Inflammation Value (PIV) assessment based on neutrophils, platelets, monocytes, and lymphocytes numeration in liquid biopsy. The authors from this study reported that the increase of PIV in MSI-H CRC patients observed 3–4 weeks after treatment with immunotherapy was an independent predictor of clinical benefit from ICI [141]. Overall, this body of research strengthens the fact that particular immune cell types may infiltrate CRC tumors regardless of the microsatellite status, and that the assessment of such cell types could be paramount for defining immunotherapeutic strategies for patients with dMMR/MSI-H tumors.

## 6. Conclusions and Future Perspectives

The emergence of immunotherapy has offered new treatment approaches for a subset of CRC patients with dMMR/MSI-H tumors. Compared to pMMR/MSS, dMMR/MSI-H CRC acquire a privileged immune landscape that can be targeted with state-of-the-art immunomodulators, showing outstanding response rates in patients with localized and metastatic disease. Despite the recent clinical advances, a proportion of patients within this subset are still refractory to the current immunotherapeutic strategies. Ongoing clinical trials are focusing on combinations of ICIs with different formats of chemotherapy and/or biological treatments to increase response rates in resistant dMMR/MSI-H CRC. 

On the other hand, the realization that a subset of pMMR/MSS CRC patients may also benefit from immunotherapy as well as the existence of MSS/MSI-het CRCs call for the need of predictive markers beyond MSS/MSI status. For instance, the discovery and clinical use of original biomarkers could help the clinical decision-making by predicting benefit from immunotherapy not only in dMMR/MSI-H CRC patients but also in pMMR/MSS and MSS/MSI-het CRCs. In this line, factors such as the expression of immunomodulating proteins, the microbiome and the composition of immune infiltrates may shed light on resistance and response mechanisms, which could be leveraged to expand the population of responsive patients treated with immunotherapy and provide a rationale for the discovery of new druggable targets.

## Figures and Tables

**Figure 1 cancers-15-01022-f001:**
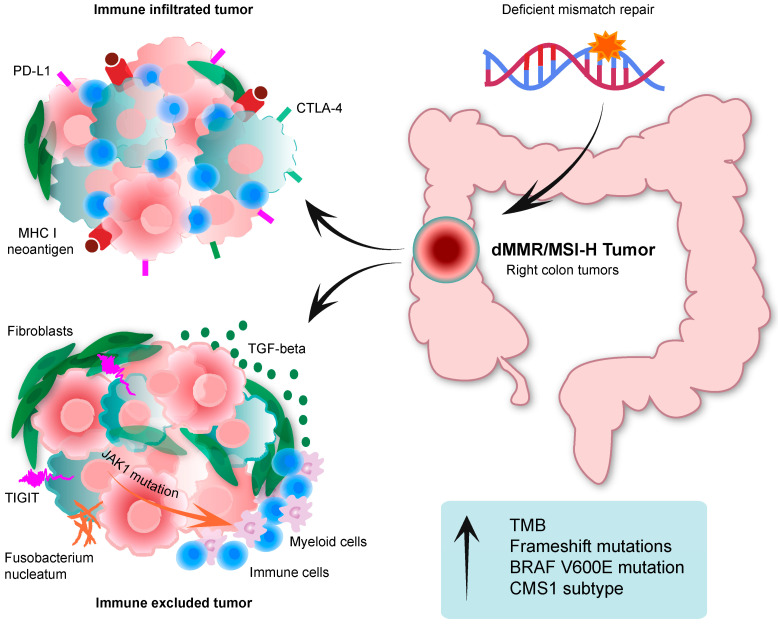
Molecular features of immune infiltrated or immune excluded dMMR/MSI-H CRC. Immune infiltrated tumors are characterized by increased PD-L1 and CTLA-4 expression as well as elevated neoantigen presentation. Immune excluded tumors feature a TGF-beta enriched tumor microenvironment with abundant immunosuppressive myeloid cells and fibroblasts. Arrow in blue box means increased. dMMR: deficient mismatch repair. MSI-H: high microsatellite instability. PD-L1: programmed death ligand 1. CTLA-4: cytotoxic T-lymphocyte antigen. MHC I: major histocompatibility complex 1. TIGIT: T cell immunoglobulin and ITIM domain. TMB: tumor mutational burden. CMS1: consensus molecular subtype 1.

**Table 1 cancers-15-01022-t001:** Clinical outcome in pivotal trials involving dMMR/MSI-H mCRC patients.

	KEYNOTE-177	Checkmate 142				Keynote 164		Keynote 016
Treatment	1st	1st	1st	>2nd	>2nd	>2nd	>3rd	>3rd
Phase	III	III	II	II	II	II	II	II
N patients	153	154	45	74	119	63	61	10
Schedule	Pembro	Chemo	Nivo + Ipi	Nivo	Nivo + Ipi	Pembro	Pembro	Pembro
ORR	44%	33%	60%	31%	55%	33%	33%	40%
DCR	65%	75%	84%	69%	80%	57%	51%	90%
mPFS	16.5 m	8.2 m	NR	14.3	NR	4.1 m	2.3 m	NR
12 m PFS	55%	37%	77%	50	71%	41%		78%
mOS			NR	NR	NR	NR	31.4 m	NR
12 m OS			83%	73	85%	76%	72%	NR

dMMR/MSI-H, deficient mismatch repair, microsatellite instability-High; Pembro, Pembrolizumab; Nivo, Nivolumab; Ipi, Ipilimumab; Chemo, chemotherapy, ORR, overall response rate; DCR, disease control rate; PFS, progression-free survival; OS, overall survival; NR, not reached.

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
