# Peer review of "Challenges and Therapeutic Opportunities in the dMMR/MSI-H Colorectal Cancer Landscape"

_cancers, 2023, doi:10.3390/cancers15041022_

Round 1
Reviewer 1 Report
Globally the manuscript review is well written and covers different hot topics in MSI colorectal tumors. The authors separate in 4 sections (clinical assessment, molecular features therapeutic implications and predictive factors) that facilitate the lecture.
Despite of it, there are several aspects that would be revised to improve it
Major comments:
I suggest to include Pan-Immune-Inflammation Value (PIV) in the section 5.3 (Molecular factors). I also suggest to include transcriptomic and metabolomic studies as a potential predictive mechanism of resistance in MSI colorectal cancer patients.
Minor comments:
Lines 63 to 66 should be re-written or deleted. It is not clear that MSI metastatic patients have more aggressive disease or respond poorly to conventional therapies (pe. control arm in KEYNOTE 177). Again lines 67 and 68 (dismal prognosis….) should be revised.
Lines 154-156. The are other mutations that are more frequently found in MSI colorectal cancer patients than BRAF and RAS mutations. It would be more appropriate if the authors want to emphasize the differences between RAS and BRAF frequencies in MSI and MSS patients.
In the neoadjuvant part 4.2, I suggest to include the NICHE2 data (although not published, it was reported in last ESMO Meeting). Also, in adjuvant ongoing clinical trials (ATOMIC and POLEM) a critical overview comparing neoadjuvant vs adjuvant strategies is lacking. Please try to add your feed-back on that.
Reviewer 2 Report
In the manuscript “Challenges and therapeutic opportunities in the dMMR/MSI-H colorectal cancer landscape” Nuria Mulet Margalef and colleagues describe the association of mismatch repair deficiency (dMMR) / microsatellite instability with the prognosis of colon cancer and treatment options. The authors very well characterize dMMR colon cancers, describe what are the causes of dMMR and microsatellite instability, how they are detected and how they affect the development of colon cancer, and what are their therapeutic implications, including predictive factors.
Unfortunately, the manuscript presents a very limited novelty, especially considering a recently published and a very similar review article by Borelli et al. (Borelli B et al. Immune-checkpoint inhibitors (ICI) in metastatic colorectal cancer (mCRC) patients beyond microsatellite instability. Cancers (Basel) (2022); 14(22):4974) and several other review articles that have been published last year (2022), for instance: (1) Yang Z et al. Current progress and future perspectives of neoadjuvant anti-PD-1/PD-L1 therapy for colorectal cancer. Front Immunol (2022); 13:1001444, (2) Taieb J et al. Deficient mismatch repair/microsatellite unstable colorectal cancer: Diagnosis, prognosis and treatment. Eur J Cancer (2022); 175:136-157 or (3) Zhang X et al. Neoadjuvant immunotherapy for MSI-H/dMMR locally advanced colorectal cancer: New strategies and unveiled opportunities. Front Immunol (2022); 13:795972.
Despite the lack of novelty, the manuscript is well-structured and the authors correctly present the current findings in the field, although I have some minor remarks that require the authors’ attention:
1) Figure 1 is somewhat misleading and hard to understand. In the Figure, the authors present the development of MSI colon cancer and its transformation (?) into an immune-excluded tumour or immune-infiltrated tumour. However, in the Section 3.2. of the manuscript, the authors write that dMMR/MSI-H tumours are “immune-infiltrated”. Also, in the Sections 3.3. and 5.3. they write that pMMR/MSS tumours are “immune-excluded”. Why then the Figure 1 shows an arrow leading from “MSI tumour” to “immune-excluded tumour”? From what the authors write in the Section 5.3., I understand that TGF-beta present in the tumour microenvironment may lead to “immune-exclusion” of dMMR/MSI-H tumours. Therefore, if there are any factors (such as TGF-beta or others) affecting such a bidirectional transformation of MSI tumours, they should be presented in the Figure. Moreover, words “immunotherapy” above the arrows suggest that this process may be triggered by immunotherapy which is misleading. In addition, please provide in the figure legend explanation of abbreviations used in the figure.
2) Throughout the whole manuscript the authors use very vague wording such as “(…) PFS and OS after a 9 months follow up were largely superior for patients with dMMR/MSI-H (…)” (Section 4.1.), “(…) reported that the early increase of PIV (…)” (Section 5.1), “(…) a majority of dMMR/MSI-H CRCs are clustered (…)” (Section 3.2) and others. Please avoid such wordings and provide precise numerical data (e.g. what exactly do you mean by “largely superior” – provide percentage/fold increase or decrease value).
3) The overall quality of English is good, however, please pay attention to certain confusing sentences that might need re-writing. (1) Simple summary: “(…) a dysfunctional DNA repair system which ultimately associates with a tumour immune microenvironment prone to respond to immunotherapy.” This sentence does not make much sense. Perhaps it would be better to write “(…) a dysfunctional DNA repair system which renders the tumour microenvironment more prone to immunotherapy.”? (2) Section 5.2 “In this line, Raymond et al. demonstrated that the gut microbiota lasts 3 months to recover from a 7-day ATB regimen. Therefore, the microbiome has emerged as a potential biomarker (…)”. Why “therefore”? There is no logical connection between these two sentences. Also, instead of “(…) lasts 3 months to recover (…)”, it would be better to write “(…) for the gut microbiota it takes 3 months to recover (…)”. (3) Section 3.1. “Importantly, BRAF V006 is unique to (…)”. Should not it be “(…) BRAF V006 mutation is unique to (…)”.
4) In several places of the manuscript the authors underscore the importance of BRAF mutations in dMMR/MSI-H tumours. Please briefly explain the role of BRAF in the text to add appropriate context.
5) When you write about therapeutic monoclonal antibodies, please present their molecular targets. In the Section 3.3., the authors do not write what are the targets of durvalumab and tremelimumab.
Round 2
Reviewer 1 Report
Now the review covers the topic adequatelly